# The Roles of *MTRR* and *MTHFR* Gene Polymorphisms in Colorectal Cancer Survival

**DOI:** 10.3390/nu14214594

**Published:** 2022-11-01

**Authors:** Yu Wang, Meizhi Du, Jillian Vallis, Matin Shariati, Patrick S. Parfrey, John R. Mclaughlin, Peizhong Peter Wang, Yun Zhu

**Affiliations:** 1Department of Epidemiology and Biostatistics, School of Public Health, Tianjin Medical University, Tianjin 300070, China; 2Division of Community Health and Humanities, Faculty of Medicine, Memorial University of Newfoundland, St. John’s, NL A1B 3V6, Canada; 3Clinical Epidemiology Unit, Faculty of Medicine, Memorial University of Newfoundland, St. John’s, NL A1B 3V6, Canada; 4Dalla Lana School of Public Health, University of Toronto, Toronto, ON M5T 3M7, Canada

**Keywords:** *MTRR*, *MTHFR*, gene polymorphism, haplotype, folate, colorectal cancer survival

## Abstract

Background: Paradoxically epidemiological data illustrate a negative relationship between dietary folate intake and colorectal cancer (CRC) risk. The occurrence and progression of CRC may be influenced by variants in some key enzyme coding genes in the folate metabolic pathway. We investigated the correlation between genetic variants in methionine synthase reductase (*MTRR*) and methylenetetrahydrofolate reductase (*MTHFR*) and CRC survival. Methods: This study used data collected from the Newfoundland Familial Colorectal Cancer Study. A total of 532 patients diagnosed with CRC for the first time from 1999 to 2003 were enrolled, and their mortality were tracked until April 2010. DNA samples were genotyped by Illumina’s integrated quantum 1 million chip. Cox models were established to assess 33 tag single-nucleotide polymorphisms in *MTRR* and *MTHFR* in relation to overall survival (OS), disease-free survival (DFS) and CRC-specific survival. Results: The *MTRR* and *MTHFR* genes were associated with DFS and CRC-specific survival in CRC patients at the gene level. After multiple comparison adjustment, *MTRR* rs1801394 A (vs. G) allele was associated with increased DFS (*p* = 0.024), while *MTHRT* rs3737966 (G vs. A), rs4846049 (T vs. G), rs1476413 (A vs. G), rs1801131 (C vs. A), rs12121543 (A vs. C), rs1801133 (C vs. T), rs4846052 (T vs. C), rs2066471 (A vs. G) and rs7533315 (T vs. C) were related to worse CRC-specific survival. Additionally, significant interactions were seen among pre-diagnostic alcohol consumption with *MTRR* rs1801394, rs3776467, rs326124, rs162040, and rs3776455, with superior OS associated with those protective variant alleles limited to patients with alcohol consumption under the median. The *MTHFR* rs3737966 (G vs. A) allele seemed to be detrimental to CRC survival only among subjects with fruit intake below the median. Conclusions: Polymorphic variants in *MTRR* and *MTHFR* genes that code for key enzymes for folate metabolism may be associated with survival in patients with CRC. The gene-CRC outcome association seems modulated by alcohol drinking and fruit intake.

## 1. Introduction

Worldwide, more than 1.9 million new cases of colorectal cancer (CRC) were reported in 2020, with a 50% death rate, making CRC the third most common malignancy in adults [1]. Folate is an essential B vitamin involved in DNA methylation, synthesis, cellular growth, and repair, aberrations of which have been implicated in various neoplasms [2,3], the most frequently reported being CRC [4,5,6]. Paradoxically epidemiological evidence suggests a dual role of folate in CRC in which moderate dietary increases before the establishment of neoplastic foci suppress the development of tumors in normal tissues, whereas supraphysiologic doses of supplementation once early lesions are established enhances tumorigenesis [7].

Polymorphisms in the genes encoding folate metabolism enzymes add an additional layer to the complexity of the association between folate and CRC. Methylenetetrahydrofolate reductase (*MTHFR*) and methionine synthase reductase (*MTRR*) are key enzymes in homocysteine and folate metabolic pathways [8]. Two common studied single nucleotide polymorphisms (SNP) of *MTHFR* are C677T (Ala222Val, rs1801133) and A1298C (Glu429Ala, rs1801131); both mutations variably reduce *MTHFR* enzyme activity [9], resulting in elevated levels of 5,10-methylenetetrahydrofolate and thymidine, thereby triggering an increase in DNA synthesis and repair [10]. Most studies [11,12,13,14,15,16,17], but not all [18,19], reported protective effects of the C677T and A1298C mutations against CRC, with odds ratios (ORs) ranging from 0.54 to 0.80 [11,12,13,14] and 0.6 to 0.8 [15,16,17], respectively, whereas studies of the A66G (rs1801394) polymorphism of *MTRR* noted its detrimental influence, with an increase in CRC risk for GG homozygotes compared to AA homozygotes [20,21,22,23,24].

In contrast to CRC risk, the prognosis value of *MTHFR* and *MTRR* mutations has been examined in minimal research [25,26,27], one of which were conducted by members of our team [27]. The findings from these studies have generally been inconsistent. Some studies have demonstrated null association [25] or poorer survival rate related to the *MTHFR* A1298C CA/CC mutation than the wild AA genotype [28]. Others, including our previous analysis based on the Newfoundland Familial Colorectal Cancer Study (hazard ratio (HR) = 1.72) [27] and a recent meta-analysis (HR = 1.85) [26], have reported a shorter overall survival in patients with *MTHFR* A1298C CC genotype compared to those with CA/AA genotypes. No relationship between *MTHFR* C677T polymorphism and CRC prognosis has been found [26,27]. However, previous research on *MTHFR* and *MTRR* has focused on limited candidate SNPs for their relevance to CRC prognosis, including our prior analysis in which only two candidate SNPs in *MTHFR* (i.e., C677T and A1298C) were evaluated; none have used a gene-wide tag SNP panels that cover the majority of all common variations in the *MTHFR* and *MTRR* genes to detect risk alleles that are associated with CRC survival. Additionally, the interactions occurring between genetics and dietary factors are seldom assessed.

Starting from our preliminary results that evidenced an association between *MTHFR* A1298C and CRC prognosis, the present study as an outgrowth further examined allelic variations in *MTHFR* and *MTRR* in relation to CRC survival using a gene-wide tag SNP approach. Effect modifications by dietary intakes of B vitamins, alcohol, and fruits were further explored.

## 2. Materials and Methods

### 2.1. Study Population

The study participants were CRC patients drawn from the Newfoundland Colorectal Cancer Registry, a resource for studies on genetic and environmental risk factors of CRC that was initiated in 1999 and is described in detail elsewhere [29,30,31]. In brief, population-based case patients with a newly diagnosed CRC from 1999 to 2003 and aged 20–75 years were eligible for inclusion. From the Newfoundland Colorectal Cancer Registry, a total of 1126 eligible patients were identified; of these, 737 consenting patients completed and returned detailed epidemiological questionnaires. A germline DNA sample and disease outcome data were available on 532 confirmed CRC patients. Informed consent was obtained from all participants. Ethical approval for this study was obtained from the Research Ethics Board of the Memorial University of Newfoundland (No. 40001511).

### 2.2. Baseline Information

Data on demographics (e.g., age, sex and race), living habits (e.g., smoke, drink status and exercise), medical history, and personal and familial history of cancer were collected from each subject through a self-administered family history questionnaires and a personal history questionnaire (the median time from date of diagnosis to study enrollment was 1.8 years). Dietary intake from one year prior to diagnosis was collected via a 169-item food frequency questionnaire, which had previously been validated in the Newfoundland population [32]. The average nutrient intakes were calculated by multiplying the frequencies of consumption of each item by the nutrient content per unit.

### 2.3. Study Outcomes

Participants in this study were followed from cancer diagnosis until April 2010. The main outcomes were overall survival (OS), defined as the time from CRC diagnosis until all-cause death; disease-free survival (DFS), defined as time from CRC diagnosis until death from any causes, CRC recurrence or metastasis, whichever came first; and CRC-specific survival, measured from the date of cancer diagnosis to death from CRC. Patients alive and free of all these events at the end of study were censored at the date of last contact.

### 2.4. SNP Genotyping and Selection

Genotyping was performed on DNA from peripheral blood via the Illumina Human Omni-Quad Bead chip that contains about 1.1 million SNPs at Centrillion Biosciences (Palo Alto, CA, USA). For quality control purpose, 200 duplicates were genotyped using the Affymetrix Axiom my Design GW Array Plate (Thermo Fisher Scientific, Waltham, MA, USA), which contains 1.3 million probes. SNPs with genotyping concordance smaller than 97% were removed from the analysis.

We used a SNP tagging approach to minimize the number of SNPs to be examined. A subset of SNPs (tag SNPs) capturing most of the common variation in the *MTRR* and *MTHFR* genes were selected using Plink v1.07 (http://www.cog-genomics.org/plink/1.9/ accessed 20 March 2022) according to following criteria: the minor allele frequency of SNP ≥ 5%; pairwise r^2^ > 0.9; and distance from any adjacent SNPs greater than 50 base pairs [33]. We selected 17 SNPs in the *MTRR* gene and 16 SNPs in the *MTHFR* gene. Several hotspot SNPs reported in previous research were additionally included (i.e., *MTRR* rs1801394, *MTHFR* rs1801131 and *MTHFR* rs1801133. Genotype distributions of all SNP were in line with the expected Hardy–Weinberg equilibrium.

### 2.5. Statistical Analysis

Comparisons of baseline variables between groups were performed with Log-Rank test. Further analysis was stratified by anatomical site of cancer.

We used a principal component analysis accounting for linkage disequilibrium (LD) among SNPs to examine overall association of a gene with CRC survival [34]. Uncorrelated linear combinations of original SNPs that explain the greatest amount of variance in the entire gene were calculated. The number of principal components retained in the Cox model was determined by 80% explained-variance threshold. A global *p*-value for an overall association between gene and CRC survival was computed via a likelihood ratio test comparing models with and without principal components with degrees of freedom equal to the total number of principal components.

Cox proportional hazard models estimated the HRs and 95% confidence intervals (CIs) for the association between individual SNPs and disease outcomes. Covariates were selected using the stepwise approach based on a *p* value less than 0.05. The final models adjusted for sex, race, age at diagnosis, disease stage at diagnosis, marital status, microsatellite instability (MSI) status, alcohol drinking and folate intake. *p* values for all SNPs were adjusted for multiple comparisons accounting for correlated SNPs with a modified test of Conneely and Boehnke [35].

For haplotype analysis, haplotype blocks and LD plots were generated by Haploview v4.2 (https://www.broadinstitute.org/haploview/downloads accessed 10 April 2022). Haplotypes for individual study subjects were assembled a “best” reconstruction from the genotyped data by PHASE v2.1 program [36]. The relationship between individual haplotypes and CRC survival was assessed by modeling all haplotypes simultaneously using the most frequent haplotype as the reference. The global *p*-value was computed for each haplotype block with a likelihood ratio test. Haplotypes with a frequency less than 0.01 and their subjects were removed from the analysis. Bonferroni correction for multiple comparisons was performed in haplotype analysis where individual tests each used a *p* value threshold of 0.05/adjusted *p* value = 0.0016.

We estimated stratum-specific HRs to evaluate potential effect modification of genetic variants by dietary factors in CRC survival. Heterogeneity of the HRs by dietary intake was evaluated by way of a two-sided Wald test for regression models that included interaction terms between SNPs and dietary factors. All analyses were performed using SAS v9.4 (SAS Institute, Cary, NC, USA) and RStudio (R Foundation for Statistical Computing, Vienna, Austria), and figures were produced with Graphpad Prism 9.0 (GraphPad Software, San Diego, CA, USA).

## 3. Results

### 3.1. Patient Baseline Characteristics

Of the 532 CRC patients enrolled in the current study, 330 (62.03%) were men, and 440 (96.92%) were from white ethnic groups (Table 1). The mean age at study enrollment among participants was 60.06 (±9.23) years. Three hundred and three (56.96%) CRC patients were diagnosed at an early stage (stage I/II), and the majority (65.83%) had cancers at colon subsite. Of the 504 patients with MSI information, 446 (88.49%) were classified as microsatellite stable or microsatellite instability-low. During follow-up (median follow-up time, 6.4 years), 183 patients died from all causes; 213 patients died from any cause or experienced a cancer recurrence or metastasis; and 94 patients died from CRC. Log-rank test indicated that sex, race, stage at diagnosis, MSI status and red meat intakes were significantly associated with overall survival.

### 3.2. Gene-Level Association with CRC Survival

Principal component analysis was applied to assess the overall gene-level association of *MTRR* and *MTHFR* with survival among CRC patients (Table 2). Although *MTRR* was not related to survival of CRC patients as a whole, significant site-specific associations were observed for the *MTRR* gene with DFS (Global *p* = 0.015) and CRC-specific survival (Global *p* = 0.025) among colon cancer patients. With respect to *MTHFR*, gene-level association was seen for CRC-specific survival in total CRC patients (Global *p* = 0.0005) and colonic cancer patients (Global *p* = 0.0004).

### 3.3. Single SNPs and CRC Survival

The associations between individual SNPs within each gene for any, colon, and rectal cancers were examined using additive models for each SNP. Of the 17 SNPs in the *MTRR* gene, the rs1801394 homozygous mutant GG genotype was associated with decreased DFS in colon cancer patients, even after the adjustment for multiple comparisons (A vs. G allele, HR = 0.60, 95%CI: 0.44, 0.81) (Appendix A and Figure 1C). Of the 16 SNPs in the *MTHFR* gene, 8 SNPs specifically, rs3737966 (G vs. A allele), rs4846049 (T vs. G allele), rs1476413 (A vs. G allele), rs1801131 (C vs. A allele), rs12121543 (A vs. C allele), rs4846052 (T vs. C allele), rs2066471 (A vs. G allele), and rs7533315 (T vs. C allele), were correlated with decreased CRC-specific survival in all cases (Appendix A and Figure 1A). Further, 7 SNPs, including the hotspot SNP in prior research, rs1801133 (Figure 1B), demonstrated some suggestion of association with colon cancer-specific mortality.

### 3.4. Associations between Haplotypes and CRC Survival

To detect possible epistatic effects, we constructed haplotype blocks formed by the markers that are in disequilibrium. A total of three major haplotype blocks on *MTRR* and four major blocks on *MTHFR* were identified (Figure 2, Appendix A). In the *MTRR* gene, a G-C haplotype in LD block 1 (rs1801394, rs13181011) was associated with a poorer OS (HR = 1.63, 95%CI: 1.08–2.48), while the haplotypes designated G-T and G-C (rs1801394, rs13181011) were associated with a reduced DFS (HR = 1.44, 95%CI: 1.04–2.01, HR = 1.68, 95%CI: 1.14–2.46, Global *p* = 0.038) for colon cancer patients. For the *MTHFR* gene, a G-A haplotype (rs4846048, rs2184226) led to worse OS in colon cancer (HR = 1.45, 95%CI: 1.01–2.06). Patients with the haplotype T-G (rs4846049 and rs1476413) were more likely to experience a reduction in OS for colon cancer patients (HR = 1.71, 95%CI: 1.00–2.91) and CRC-specific survival (HR = 2.73, 95%CI: 1.30–5.74). A C-A haplotype in block 3 (rs1801131 and rs12121543) of *MTHFR* was related to an increased odds of all-cause death of colon cancer (HR = 1.50, 95%CI: 1.05–2.13) and CRC-specific death (HR = 1.54, 95%CI: 1.01–2.35), while a C-C haplotype (rs1801131 and rs12121543) was associated with a higher risk of CRC-specific mortality (HR = 2.18, 95%CI: 1.06–4.51). However, none of the above results were significant after adjusting for multiple comparisons.

### 3.5. Gene-Diet Interactions

To evaluate the possibility of gene-diet interactions, participants were grouped as high versus low intakes of vitamin B, fruits and alcohol based on the respective median value (Figure 3 and Figure 4). We observed significant interactions among pre-diagnostic alcohol consumption with several SNPs in *MTRR* (i.e., rs1801394, rs3776467, rs326124, rs162040, and rs3776455), with increased OS time associated with those protective variant alleles limited to patients consuming alcohol below the median. In addition, the *MTHFR* rs3737966 (G vs. A) allele seemed to be detrimental to CRC survival only among subjects with fruit intake below the median (*p* = 0.041). No significant interactions were detected between dietary vitamin B and any of these SNPs examined in this analysis.

## 4. Discussion

In the current work, *MTRR* and *MTHFR* genes were related to DFS and CRC-specific survival at the gene level. The *MTRR* rs1801394 (A66G) was associated with reduced DFS, while in the *MTHRT* gene, rs3737966 (G vs. A), rs4846049 (T vs. G), rs1476413 (A vs. G), rs1801131 (C vs. A), rs12121543 (A vs. C), rs1801133 (C vs. T), rs4846052 (T vs. C), rs2066471 (A vs. G) and rs7533315 (T vs. C) were related to worse CRC-specific survival. Potential effect modifications were observed between *MTRR* and alcohol drinking, and between *MTHFR* and fruit intake.

The direction of our estimate that *MTRR* rs1801394 (A66G) conferred detrimental effect on DFS is concordant with the findings of many observational studies [21,24] and two meta-analyses [37,38] employing CRC incidence as an outcome. The *MTRR* A66G variant, located in 5p15.31, harbors a missense mutation arising from a 66 A-to-G substitution that changes isoleucine to methionine at *MTRR* position 22. The derived variant was predicted to influence splicing and transcriptional regulation, and thus, increase homocysteine concentrations [39,40]. Elevated homocysteine levels are thought to induce a higher risk for colorectal polyps [41]. This might help explain the allelic association between rs1801394 and DFS observed in this study. It should be noted that results from a meta-analysis by ethnicity demonstrated no significant relationship between the rs1801394 and CRC susceptibility among Asian, Caucasians, Japanese, and mixed populations [42]. However, three out of five meta-analysis [38,43,44,45,46] reported an increased risk of CRC development for *MTRR* G allele and/or GG genotype in Caucasians [38,44] and in Asians [46], suggesting a race-specific effect of *MTRR* rs1801394 on CRC. Nevertheless, the majority of participants in this study were white (96.92%), and therefore we were unable to assess the possibility of different patterns of effects across race. Further research is required to decipher the race-specific effect of *MTRR* rs1801394 on CRC prognosis. With respect to the *MTHFR* gene, two non-synonymous *MTHFR* variants, rs1801133 (C677T) and rs1801131 (A1298C) are among the most studied genetic markers for folate metabolism-related health outcomes. The C677T, resulting in a substitution of the amino acid alanine by valine, and the A1298C, resulting in a substitution of glutamate by alanine [44], were in high LD and were related to a reduced enzyme activity, an elevated level of homocysteine and a lower level of plasma folate [20,47]. Our previous analysis based on the same population using OS and DFS as outcomes [27] has reported a poorer OS in patients with *MTHFR* A1298C CC genotype compared to those with CA/AA genotypes (HR = 1.72), whereas no relationship of *MTHFR* C677T with either DFS or OS has been found. As an extension, this study additionally assessed these variants in relation to CRC-specific survival and, intriguingly, *MTHFR* 677T polymorphism was related to prolonged colon cancer-specific survival. These results are consistent with most previous reports [21,22,48] indicating a protective role of C677T against CRC.

In addition to the two commonly analyzed SNPs, our study identified novel SNPs that might influence survival after a diagnosis of CRC, including *MTHFR* rs4846049, rs3737966, rs1476413, rs12121543, rs4846052, rs2066471, rs7533315 and rs7553194. The rs4846049 and rs3737966 polymorphisms exist on the 3′UTR of the *MTHFR* gene while the others are within *MTHFR* introns. PCR analysis revealed that subjects carrying a GG genotype of rs4846049 exhibited reduced *MTHFR* gene expression [49]. Current research on *MTHFR* rs4846049 polymorphism mainly focused on diseases such as acute lymphoblastic leukemia and preeclampsia [49,50]. This is the first study linking *MTHFR* rs4846049 G > T to poorer survival among CRC patients; the results are in agreement with the only few studies published on CRC risk so far, in which *MTHFR* rs4846049 CA + AA vs. CC was associated with a modest, significantly increased risk for this cancer [51]. Similarly, *MTHFR* rs3737966 has not yet been linked to CRC prognosis, but this variant exists in the miRNA binding site of *MTHFR* and might interfere with the binding of miRNA to target mRNA [52]. The mechanisms underlying the associations between some intronic variants and CRC survival remain undetermined, but intronic variants have been suggested to affect alternative splicing by interfering with splice site recognition [53].

For the *MTRR* gene, the relationship between haplotype G-C/G-T, composed of rs1801394 (exon 2) and rs13181011 (intron 3), and the worse survival of colon cancer patients provides good evidence for a harmful effect of the rs1801394 G allele on survival. For the *MTHFR* gene, haplotypes T-G, consisting of rs4846049 and rs1476413, C-A and C-C, consisting of rs1801131 (exon 8) and rs12121543 (intron 7), were associated with a significant reduction in CRC-specific survival. This outcome appears logical and consistent with the results of the genotype analysis for individual SNPs in the two genes. We speculate that these SNPs or those located in close proximity in the genome may harbor causal variants that in conjunction with each other affect health outcome; although not all functional, these SNPs can be considered as candidate markers for future association studies to detect health-related genetic variants.

One of the novel findings of our study is the gene-diet interactions between gene polymorphisms and CRC survival. *MTRR* gene polymorphisms conferred a protective effect on survival in the low alcohol drinking group, suggesting that alcohol consumption may be a modulator for the gene-survival relationship. High alcohol intake can be regarded a low-methyl diet [54] while *MTRR* may affect the methylation process by leading to the decreased activity of methionine synthase [46], which may jointly give rise to the risk of CRC. Nevertheless, no interaction was found between the *MTRR* gene and alcohol consumption in relation to the risk of colorectal adenoma in a Japanese study [55], possibly due to differences in ethnicity and gender of the study populations (e.g., only Asian men were investigated in that Japanese study). The *MTHFR* rs3737966 G allele was related to decreased survival only among the group with low fruit intake. If present results are validated in further research, with extended sample size, then CRC patients, especially those with unfavorable genotypes, may receive survival benefit through increasing fruit consumption.

The strengths of this study include the relatively large sample size, the long follow-up period of up to 10 years, and comprehensive gene coverage (i.e., a gene-wide tag SNP panels that cover the majority of common variations in the *MTHFR* and *MTRR* genes were studied). There are some limitations. First, some rare haplotypes may result in lower power for part of the analyses. Second, exposure to environmental factors in the cohort participants were assessed at the time of cancer diagnosis, and thus, we were unable to examine how health behavior changes following cancer diagnosis could modulate the way genes work. This provides the impetus for future replication studies using post-diagnosis exposures.

## 5. Conclusions

Our data demonstrated that genetic variation in *MTRR* and *MTHFR*, as measured by a tag SNP approach, seemed to play an independent role in CRC survival. The gene-CRC outcome association was modulated by alcohol drinking. Our analysis highlights the concrete value, as prognostic value, of the *MTRR* and *MTHFR* gene variants in CRC patients and sheds additional light on potentially modifiable factors that could be targeted to improve prognosis in CRC survivors.

## Figures and Tables

**Figure 1 nutrients-14-04594-f001:**
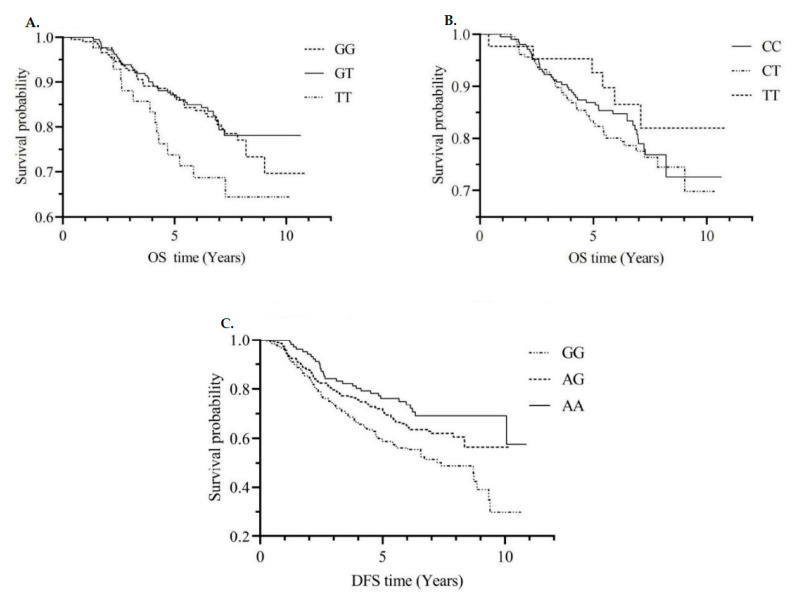
CRC-specific survival curves by *MTHFR* rs4846049 genotype (**A**); CRC-specific survival curves by *MTHFR* rs1801133 genotype (**B**); DFS curves by *MTRR* rs1801394 genotype (**C**). Abbreviations: CRC = colorectal cancer; *MTHFR* = methylenetetrahydrofolate reductase; OS = overall survival; DFS = disease-free survival. GG, GT and TT; CC, CT and TT; GG, AG and AA are genotypes of SNPs.

**Figure 2 nutrients-14-04594-f002:**
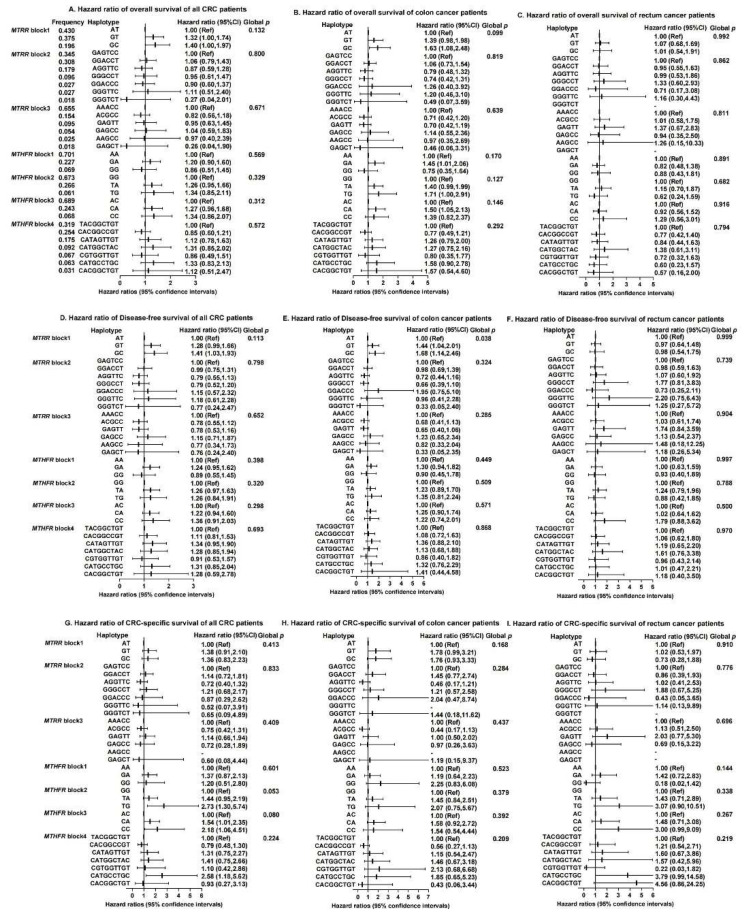
Haplotypes on *MTRR* and *MTHFR* genes and associations with overall survival, disease-free survival and CRC-specific survival among colorectal cancer patients (**A**–**I**). Abbreviations: CRC = colorectal cancer; *MTRR* = methionine synthase reductase; *MTHFR* = methylenetetrahydrofolate reductase; Ref = Reference; CI = confidence interval. *MTRR*, block1 includes rs1801394 and rs13181011. *MTRR*, block2 includes rs326124, rs1532268, rs7703033, rs6555501, rs162031 and rs162033. *MTRR*, block3 includes rs161871, rs162040, rs3776455, rs10380 and rs9332. *MTHFR*, block1 includes rs4846048 and rs2184226. *MTHFR*, block2 includes rs4846049 and rs1476413. *MTHFR*, block3 includes rs1801131 and rs12121543. *MTHFR*, block4 includes rs1801133, rs1572151, rs4846052, rs2066471, rs13306567, rs7533315, rs9651118, rs7553194 and rs13306561. Haplotype frequency is calculated based on the *n* = 532 sample. Cox proportional hazard models were adjusted for age at diagnosis, sex, stage at diagnosis, race, drink status, body mass index (BMI), screen status, BRAF2, microsatellite instability (MSI) status and marital status, where appropriate.

**Figure 3 nutrients-14-04594-f003:**
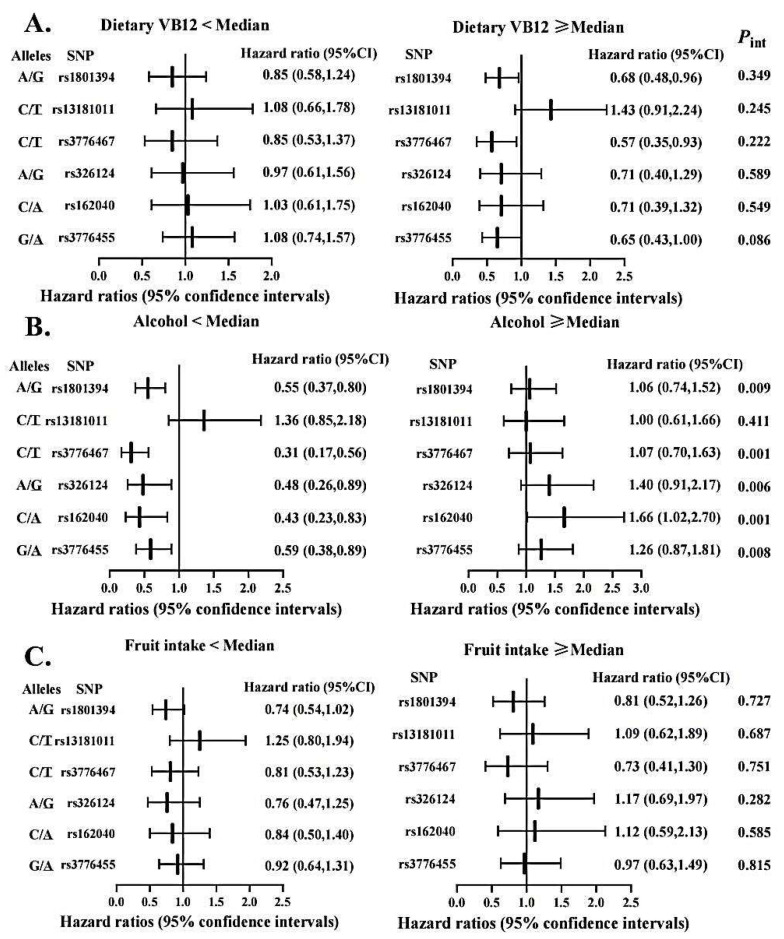
Association between selected genetic variations in *MTRR* and CRC overall survival stratified by dietary vitamin B12 (**A**), alcohol intake (**B**), and fruit consumption (**C**). Abbreviations: SNP = single nucleotide polymorphisms; VB12 = vitamin B12. Two variants at the locus presented as: Minor/Major allele. Hazard ratios were calculated in reference to the allele underlined. Cox proportional hazard models were age at diagnosis, sex, stage at diagnosis, race, BMI, marital status and MSI status, where appropriate. The medians of dietary vitamin B12, alcohol drinking, and fruit intake were 7.30 µg/day, 2.17 g/day, and 7 pieces/week, respectively.

**Figure 4 nutrients-14-04594-f004:**
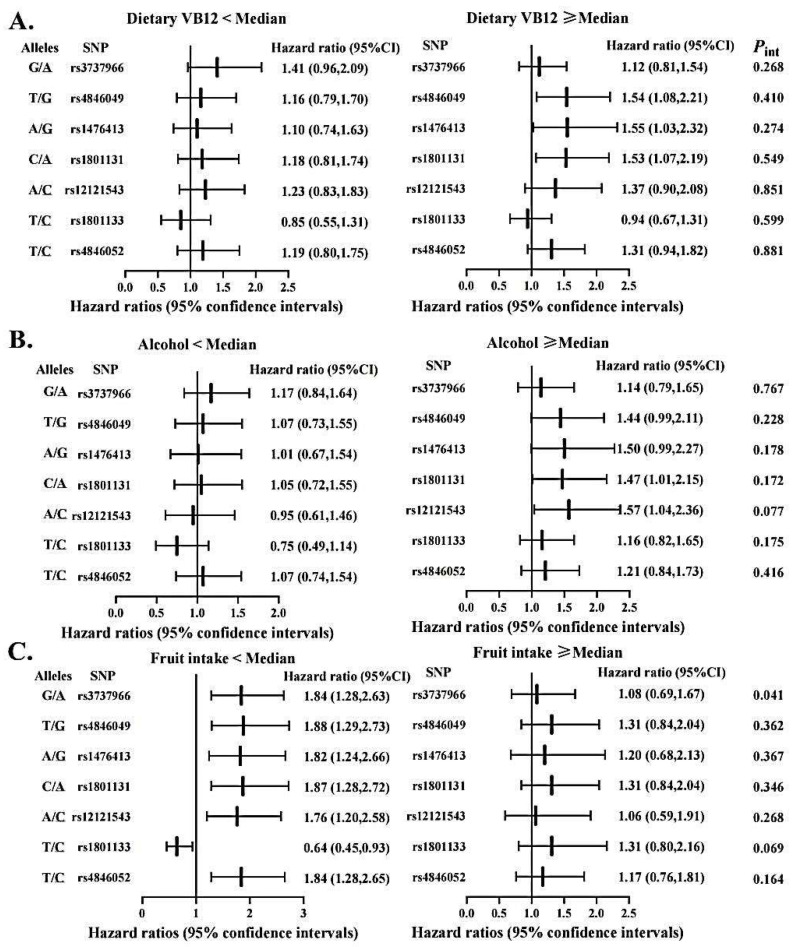
Association between selected genetic variations in *MTHFR* and CRC overall survival stratified by dietary vitamin B12 (**A**), alcohol intake (**B**), and fruit consumption (**C**). Abbreviations: SNP = single nucleotide polymorphisms. Two variants at the locus presented as: Minor/Major allele. Hazard ratios were calculated in reference to the allele underlined. Cox proportional hazard models were adjusted for age at diagnosis, sex, stage at diagnosis, race, BMI, BRAF2, smoke status, marital status and MSI status, where appropriate. The medians of dietary vitamin B12, alcohol drinking, and fruit intake were 7.30 µg/day, 2.17 g/day, and 7 pieces/week, respectively.

**Table 1 nutrients-14-04594-t001:** Baseline characteristics of patients in the Newfoundland Familial Colorectal Cancer Study (NFCCS).

Characteristic	No. Patients (%) ^a^	No. Deaths (%) ^a^	MST (Years)	*p_log-rank_*
Age at diagnosis (years) ^b^	60.06 (±9.23)	60.66 (±9.76)	-	-
Sex				0.005
Male	330 (62.03)	127 (69.40)	6.28	
Female	202 (37.97)	56 (30.60)	6.53	
Race				0.009
White	440 (96.92)	133 (94.33)	6.44	
Other	14 (3.08)	8 (5.67)	4.66	
Stage at diagnosis				<0.001
Ⅰ	94 (17.67)	18 (9.84)	6.41	
Ⅱ	209 (39.29)	58 (31.69)	6.64	
Ⅲ	178 (33.46)	65 (35.52)	6.42	
Ⅳ	51 (9.59)	42 (22.95)	3.90	
Tumour location				0.444
Colon	341 (65.83)	110 (62.86)	6.40	
Rectum	177 (34.17)	65 (37.14)	6.33	
MSI status				<0.001
MSS/MSI-L	446 (88.49)	168 (96.55)	6.33	
MSI-H	58 (11.51)	6 (3.45)	6.67	
Reported screening procedure				0.059
Yes	52 (11.45)	10 (7.09)	6.57	
No	402 (88.55)	131 (92.91)	6.40	
BMI (kg/m^2^)	8 (1.60)			0.097
<18.5	138 (27.60)	6 (3.57)	4.69	
18.5–24.9	205 (41.00)	41 (24.40)	6.40	
25.0–29.9	149 (29.80)	74 (44.05)	6.40	
≥30.0		47 (27.98)	6.33	
Average drinks per week				0.062
0.0	170 (39.44)	46 (34.59)	6.53	
0.0–6.9	138 (32.02)	43 (32.33)	6.42	
7.0–14.0	74 (17.17)	23 (17.29)	6.44	
>14.0	49 (11.37)	21 (15.79)	5.86	
Smoking status				0.133
Yes	375 (72.39)	136 (77.71)	6.36	
No	143 (27.61)	39 (22.29)	6.41	
Red meat intake (servings/week)				0.048
≤2.0	84 (16.47)	25 (14.71)	6.72	
2.1–3.0	257 (50.39)	86 (50.59)	6.42	
3.1–5.0	83 (16.27)	34 (20.00)	6.15	
>5.0	86 (16.86)	25 (14.71)	6.27	
Ever taken folate regularly				0.236
Yes	422 (93.99)	128 (92.09)	6.42	
No	27 (6.01)	11 (7.91)	6.32	
Vitamin B12 (µg/day)	9.03 (±6.06)	9.58 (±5.50)	-	-

Abbreviations: No. = number; BMI = body mass index; MSI = microsatellite instability; MST = median overall survival time; MSI-H = microsatellite instability-high; MSS/MSI-L = microsatellite stable/microsatellite instability-low. **^a^** Patients were diagnosed with colorectal cancer for the first time from 1999 to 2003 and followed for recurrence and mortality until April 2010. ^b^ Continuous variables presented as Mean ± SD (standard deviation).

**Table 2 nutrients-14-04594-t002:** Associations between *MTRR* and *MTHFR* genes and colorectal cancer overall, disease-free and CRC-specific survival.

	Overall Survival HR (95% CI) ^a^			Disease-Free Survival HR (95% CI) ^a^			CRC-Specific Survival HR (95% CI) ^b^		
	All	Colon	Rectum	All	Colon	Rectum	All	Colon	Rectum
*MTRR*									
PC1	0.94 (0.77, 1.14)	0.82 (0.64, 1.07)	1.11 (0.79, 1.55)	0.83 (0.69, 1.00)	0.72 (0.56, 0.92)	1.02 (0.74, 1.39)	0.85 (0.63, 1.15)	0.62 (0.37, 1.02)	1.03 (0.66, 1.59)
PC2	1.00 (0.83, 1.19)	0.91 (0.72, 1.14)	1.20 (0.86, 1.66)	0.96 (0.82, 1.14)	0.93 (0.75, 1.15)	1.07 (0.80, 1.44)	1.01 (0.78, 1.31)	1.19 (0.81, 1.73)	1.05 (0.62, 1.80)
PC3	0.85 (0.70, 1.04)	0.78 (0.61, 1.00)	1.06 (0.72, 1.55)	0.90 (0.75, 1.08)	0.82 (0.66, 1.03)	1.19 (0.85, 1.66)	1.06 (0.84, 1.36)	0.93 (0.67, 1.31)	1.29 (0.81, 2.04)
PC4	0.92 (0.75, 1.11)	0.84 (0.64, 1.09)	1.01 (0.71, 1.44)	0.91 (0.77, 1.09)	0.84 (0.66, 1.07)	0.94 (0.70, 1.27)	0.83 (0.63, 1.10)	0.57 (0.37, 0.87)	1.01 (0.63, 1.62)
Global *p*	0.414	0.082	0.794	0.149	0.015	0.843	0.533	0.025	0.863
*MTHFR*									
PC1	1.22 (1.02, 1.46)	1.23 (0.98, 1.55)	1.17 (0.86, 1.59)	1.13 (0.96, 1.33)	1.16 (0.94, 1.43)	1.05 (0.79, 1.40)	1.78 (1.37, 2.31)	2.08 (1.40, 3.08)	1.64 (1.09, 2.48)
PC2	0.92 (0.76, 1.11)	0.86 (0.64, 1.15)	0.84 (0.62, 1.15)	0.93 (0.78, 1.11)	0.92 (0.71, 1.18)	0.96 (0.73, 1.27)	1.01 (0.79, 1.29)	0.98 (0.61, 1.59)	0.88 (0.60, 1.27)
PC3	1.11 (0.92, 1.33)	1.08 (0.86, 1.35)	1.32 (0.92, 1.90)	1.09 (0.92, 1.29)	1.08 (0.89, 1.32)	1.13 (0.81, 1.58)	1.21 (0.96, 1.52)	1.41 (1.05, 1.89)	1.08 (0.70, 1.66)
PC4	1.03 (0.88, 1.20)	1.14 (0.93, 1.39)	0.85 (0.62, 1.16)	1.04 (0.90, 1.20)	1.11 (0.91, 1.35)	0.96 (0.74, 1.24)	0.93 (0.75, 1.15)	1.19 (0.89, 1.59)	0.55 (0.31, 0.96)
PC5	0.96 (0.79, 1.16)	0.99 (0.78, 1.26)	0.77 (0.50, 1.19)	1.04 (0.88, 1.24)	1.05 (0.85, 1.30)	0.94 (0.66, 1.35)	1.32 (0.96, 1.81)	1.60 (1.04, 2.47)	0.93 (0.49, 1.78)
Global *p*	0.226	0.233	0.291	0.511	0.493	0.976	0.0005	0.0004	0.051

Abbreviations: CRC = colorectal cancer; HR = hazard ratio; CI = confidence interval; *MTRR* = methionine synthase reductase; *MTHFR* = methylenetetrahydrofolate reductase; PC = principal component. **^a^** Cox proportional hazard models were adjusted for age at diagnosis, sex, stage at diagnosis, race, drink status, MSI status and marital status. **^b^** Cox proportional hazard models for CRC-specific survival were adjusted for age at diagnosis, sex, stage at diagnosis, drink status, MSI status and marital status.

## Data Availability

Data are available upon reasonable request.

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
