# Peer review of "The Roles of MTRR and MTHFR Gene Polymorphisms in Colorectal Cancer Survival"

_nutrients, 2022, doi:10.3390/nu14214594_

Round 1

Reviewer 1 Report

Wang et al. presented the roles of MTRR and MTHFR gene polymorphisms in colorectal cancer survival in this manuscript. Using the data from Newfoundland Familial Colorectal Cancer Study, cox models have been used to assess SNPs in MTRR and MTHFR in relation to overall survival (OS), disease-free survival (DFS), and CRC-specific survival.

Polymorphic variants in MTRR and MTHFR genes encoding for enzymes of folate metabolism have been associated with CRC patients’ survival. The genes and CRC outcome association are shown to be influenced by alcohol and fruit intake.

Several technical issues should be addressed before publication.

1.      Table 2 summarizing the principle component analysis should be presented in the form of a PCA plot or some other related figure. The table should be moved to source or supplementary data.

2.      Tables 3 and 4 hazard ratios should be represented in the form of forest plots as it is difficult to comprehend the information presented.

Author Response

Reviewer 1_1. Table 2 summarizing the principle component analysis should be presented in the form of a PCA plot or some other related figure. The table should be moved to source or supplementary data.

Response: We thank the reviewer for this comment. In table 2, the global P-value is the most important summary of the overall gene-level association between MTRR, MTHFR, and CRC survival. A table with global P-values is still our preference to represent the data. We hope this is acceptable.

Reviewer 1_2. Tables 3 and 4 hazard ratios should be represented in the form of forest plots as it is difficult to comprehend the information presented.

Response: We agree and have visualized the hazard ratios in Tables 3 and 4 with forest plots as suggested.

Reviewer 2 Report

Discussion needs to be shortened 

Author Response

Reviewer 2_1. Discussion needs to be shortened 

Response: Thanks. The discussion section has been shortened based on the reviewer’s suggestion.

Reviewer 3 Report

Please see report uploaded
